# How Rearing Systems for Various Species of Flies Benefit Humanity

**DOI:** 10.3390/insects14060553

**Published:** 2023-06-14

**Authors:** Carlos Pascacio-Villafán, Allen Carson Cohen

**Affiliations:** 1Red de Manejo Biorracional de Plagas y Vectores, Clúster Científico y Tecnológico BioMimic®, Instituto de Ecología A.C., Xalapa 91073, Veracruz, Mexico; 2Insect Rearing Education and Research, Department of Entomology & Plant Pathology, NC State University, Raleigh, NC 27695, USA

**Keywords:** Diptera rearing, human wellbeing, animal feed, medical maggots, pollination services, forensic entomology

## Abstract

**Simple Summary:**

Within the fly family (Diptera), various species transmit diseases to humans and farm animals or are pests of many crop plants. Most species of flies, however, play key ecological roles in sustaining life on Earth, and several species are reared at different scales and for various beneficial purposes worldwide. Here, we review the special place of fly rearing in relation to the tremendous role that Diptera rearing technology has contributed to the development of our state of knowledge of genetics, sterile insect technique (SIT), biological control, and quality control. We summarize information on the rearing of flies in the fields of the animal feed and human food industries, pollination services, pest control, medical wound therapy treatments, criminal investigations, and as model organisms for the development of several biological and medical fields. We document the importance of fly rearing technology as a basis for many other rearing systems approaches, making the history of Diptera rearing as a launching point for most other insect rearing systems.

**Abstract:**

Flies (Diptera) have played a prominent role in human history, and several fly species are reared at different scales and for different beneficial purposes worldwide. Here, we review the historical importance of fly rearing as a foundation for insect rearing science and technology and synthesize information on the uses and rearing diets of more than 50 fly species in the families Asilidae, Calliphoridae, Coelopidae, Drosophilidae, Ephydridae, Muscidae, Sarcophagidae, Stratiomyidae, Syrphidae, Tachinidae, Tephritidae, and Tipulidae. We report more than 10 uses and applications of reared flies to the well-being and progress of humanity. We focus on the fields of animal feed and human food products, pest control and pollination services, medical wound therapy treatments, criminal investigations, and on the development of several branches of biology using flies as model organisms. We highlight the relevance of laboratory-reared *Drosophila melanogaster* Meigen as a vehicle of great scientific discoveries that have shaped our understanding of many biological systems, including the genetic basis of heredity and of terrible diseases such as cancer. We point out key areas of fly-rearing research such as nutrition, physiology, anatomy/morphology, genetics, genetic pest management, cryopreservation, and ecology. We conclude that fly rearing is an activity with great benefits for human well-being and should be promoted for future advancement in diverse and innovative methods of improving existing and emerging problems to humanity.

## 1. Introduction

True flies (Diptera) are one of the largest and most diverse groups of creatures on Earth, living in various habitats, where they exploit virtually any type of niche [1,2,3,4]. Flies are holometabolous, with larvae and adults showing distinct and often contrasting life histories and feeding habits [1]. Adult flies have one pair of true wings, and their mouthparts are specialized for lapping/sucking liquids [1]. In some species, adults feed on nectar and can be important plant pollinators [5], others are parasites that feed on the blood of mammals, including humans, whereas adults do not feed on other species [6]. The larvae lack segmented legs and have well-developed mouthparts with chewing mandibles or mouth hooks [1,7]. Larvae can be phytophagous, saprophagous, carnivorous, fungivores, hematophagous, or coprophagous and thus act as decomposers, predators, parasites, parasitoids, or pests [6].

Flies and humans have coexisted since time immemorial, and they helped shape the history of humankind [8]. According to biblical accounts, stable flies (*Stomoxys calcitrans* (L.) (Muscidae)) and tabanids were among the plagues that hit ancient Egypt, causing cutaneous anthrax and glanders to livestock and humans [9]. During World War II, house flies, *Musca domestica* L. (Muscidae), were used by the Japanese army as a weapon to deliver cholera into Chinese cities, sickening and killing thousands of people [9]. Fly species in the families Muscidae, Sarcophagidae, Calliphoridae, and Glossinidae, among others, transmit diseases and cause health-related problems that kill humans and other vertebrate species worldwide [10]. In Africa, it is argued that tsetse flies, *Glossina* spp. (Glossinidae), the vector of trypanosomes that cause trypanosomiasis, have historically limited local farmers, preventing economic growth in many countries [11]. Around the globe, tephritid fruit flies are among the most important economic pests of fruit and vegetable industries [12]. On the positive side, more than half of the species of flies play key roles in sustaining life on Earth [1,13]. Without flies, the pollination of several crop plant species would be undermined, risking seed and fruit production [14,15]; without the recycling services flies provide, filth and corpses would accumulate around the world [13]. From an applied perspective, maggots of necrophagous calliphorid species have been used from the Middle Ages to the present day in medical therapies to treat various chronic wounds such as diabetic foot ulcers [16,17,18]. As human food, flies are among the most widely consumed group of insects worldwide and constitute a valuable food product in various cultures [19]. The large-scale rearing of species such as the black soldier fly *Hermetia illucens* (L.) (Stratiomyidae) and *M. domestica* has emerged as a lucrative business model for transforming low-value organic waste into high-quality protein for animal feed, within a framework of circular economy [20,21,22]. These are just a few examples to illustrate the past, present, and future relevance of flies in the well-being of humanity.

Given the close association between flies and humans, and the relevance these insects have in many aspects of our lives, it is not surprising that the first insect species reared on an artificial diet (i.e., a human-made food synthesized from one or more ingredients with different level of chemical purity) was the blowfly *Calliphora vomitoria* L. (Calliphoridae), followed by *Drosophila* spp. (Drosophilidae), more than a century ago [23,24]. Ever since that rearing breakthrough, various fly species have been incorporated into the list of insects that have been or are currently reared in laboratories, factories, or farms. Here, we argue that the rearing of flies is an activity with great and diverse benefits for humanity and thus should be continuously supported and promoted by science funding agencies, governments, and private initiatives. To support our claim, we review the historical importance of fly rearing as a cornerstone of the science and technology of insect rearing, and we synthesize information on more than 10 uses, for which more than 50 selected fly species are reared. We focus on the role of reared flies in the fields of animal feed and human food, pollination services, pest control, medicinal wound treatments, forensic research, and on the use of flies as model organisms in biological and medical research. We hope that we convey the fascination and value of fly rearing so that future generations of entomologists are motivated to serve humanity by becoming Diptera-rearing specialists.

## 2. The Historical Importance of Fly Rearing as a Foundation for Insect Rearing

Diptera rearing holds a special place in the discipline of insect rearing. Although rearing silkworms clearly holds historical precedence over all other insects, with a 5000-year-old standing [25], flies are, arguably, the research subjects that deliver most of the basic current practices of insect rearing. The two oldest papers that led to the concept of artificial diets for insects were two flesh fly papers by Bogdanow [26,27]. Importantly, these early papers by Bogdanow introduced the use of bovine-derived casein, purified salt solution, and various meridic nutrients such as egg albumin. These papers even included the questions of how microbe–fly interactions played a role in the fitness of the rearing subjects—flesh flies (*Calliphora* spp.). Soon after Bogdanow established the use of artificial diet components and research on microbe–fly interactions, Delcourt and Guyenot [28] spelled out in remarkable detail the concept of controlling the rearing conditions as a prerequisite to fitness. This paper was followed by a series of other papers by Guyenot [29,30,31,32,33] and some remarkable papers by Baumberger [34,35]. These papers made significant advances in *Drosophila* dietetics (i.e., the practical application of diet development), as well as advances that transcend *D. melanogaster* Meigen rearing systems leading to the standard use of salt mixtures, basic nutrients such as flour and meal from seeds (especially corn and wheat products), and various proteins such as casein and egg albumins. Other features of the control of insect rearing systems include defining standards of environmental control such as with precision incubators [36,37], containerization [36], and attention to diet properties such as pH [38]. Notably, in the early days of diet development, the scientific community was in its “Golden Age” of microbiology and genetics, and many early studies on insects reflected the then-current interest in the role of microbes in insect nutrition, leading to numerous studies of the essentiality of various species of microbes (e.g., [34,35,39,40,41,42,43]). The works of Loeb [39] and Loeb and Northrop [42] probed the basic nutrition and physiology questions regarding temperature effects on insects, and they helped set the scene for mechanistic treatment of insect nutrients and factors other than nutrients’ effects on the metabolism of *Drosophila* and other insects that showed similar biochemical and physiological responses to complex environments. Some of these works set the stage for what we are calling scientifically based exploration of insect ecology, metabolism, insect–microbial relationships, development, and genetics—including inquiries that set the stage for rearing-related biological outcomes of culture conditions.

The establishment of drosophilid rearing techniques, including artificial diets, influenced and helped establish rearing techniques for economically important Tephritidae, including early works by Marucci and Clancy [44], Moore [45], Hagen et al. [46], Santas [47], Economopoulos and Tzanakakis [48], and Orphanidis et al. [49]. In parallel with and influenced by rearing progress in drosophilids and tephritids, screwworm rearing and the sterile insect technique (SIT) were part of the interchange of rearing ideas as exemplified by Baumhover et al. [50] and in well-known publications on screwworms such as Bushland et al. [51] and Melvin and Bushland [52,53]. In these pioneering works, the pathway from small-scale rearing (hundreds of insects per week) evolved and proceeded to millions of reared insects per day (true mass rearing) through studies that helped link the basic biology of the target insect with painstakingly development and creative application of containerization, diet development/improvement, environmental condition development, and microbial control. In accordance with these innovations and improvements, the genetic basis of SIT and basic insect genetics were advanced. It is important to recognize that the vast developments in our understanding of basic genetic concepts were derived from and wedded to the improvements in rearing concepts. The great insect geneticist Calvin Bridges made this point clear when he stated, “Plough’s work on the effect of temperature extremes on crossing-over showed that it was necessary to maintain the temperature constant for genetic reasons as well as to improve culture conditions” [37]. Bridges referred here to the works of H.H. Plough on chromosomal crossing over. In several other places in his writing about rearing *D. melanogaster*, Bridges discussed the importance of having as much control as possible in the rearing procedures to assure that the conclusions about genetics were truly reading what the genes were telling us rather than vagaries of environmental conditions. Bridges and Darby [38] expressed it eloquently, pointing out the rationale for their extensive attention to rearing conditions to bridge the gap between what geneticists read as phenotypes and the genes behind the phenotypic expression. They further pointed out that many of the traits whose genetic nature they were attempting to study could be masked or not detected due to the higher mortality of some of the genetic variants (such as various eye-color traits or numbers and placement of bristles). The authors explained that only with the most robust rearing conditions would there be adequate survival of the “weaker” genotypes that could be detected. Therefore, it could be said that the refinements of various rearing parameters were essential to getting truer or more reliable answers to the genetics inquiries being carried out, and the better the culture technique, the better the genetic science to follow the rearing improvements and refinements.

The subject of insect rearing system development in relation to studies of species of Diptera (drosophilids, tephritids, and calliphorids) was treated in depth by Cohen [25]; briefly, these were topics of artificial diets, environmental control, containerization, microbial control, and other subjects developed from the efforts to use controlled rearing practices to allow for a reliable system of genetic analysis. A highly important aspect of successful, reliable rearing systems was the application of quality control (QC), as introduced by Boller and Chambers [54], expanded by Leppla et al. [55], and reviewed by Cohen [25]. The application of statistically based QC standards and practices revolutionized all aspects of mass insect rearing. Finally, the topic of relationships between insects and their microbial associates had been a topic of intense interest and elegant investigation in the early decades of the 20th century; this topic was especially well-documented in the works of Baumberger [35] and by Michelbacher et al. [43]. It was especially impressive that these researchers were asking questions that today have regained extensive interest of the biology community, and, with the current tools of molecular genetics, these topics can be investigated with greater precision, especially regarding the fastidious microbes that cannot be cultured.

## 3. Uses and Rearing Diets of Selected Fly Species

We reported on more than 10 uses for rearing 53 fly species from 12 families (Table A1, Figure 1), including diets for these species. The families covered are (in alphabetical order) Asilidae, Calliphoridae, Coelopidae, Drosophilidae, Ephydridae, Muscidae, Sarcophagidae, Stratiomyidae, Syrphidae, Tachinidae, Tephritidae, and Tipulidae. The selection of fly species and families was based on having examples of flies reared for the bioconversion of different waste products as animal feed or human food ingredients; as a source of compounds for cosmetic applications or for biodiesel production; for the application of SIT, as hosts for parasitoid production, as parasitoids for biological control of pests, for crop pollination; as medicinal larvae in wound healing (maggot therapy) or in forensic research to determine the postmortem interval of corpses as well as evidence to solve criminal cases; and as model organisms in various fields of basic and applied research (Table A1). Table A1 does not represent an exhaustive list of all the species of flies that were reared from each of the reported families, but we do include key species on which rearing programs or rearing efforts had been implemented in the topics mentioned above.

Many species of flies are reared for different purposes, and, therefore, they can play multiple functions (Table A1). For example, *Chrysomya megacephala* (Fabricius) (Calliphoridae) may be used in forensic research, for pollination services in mango crops, waste bioconversion, and as a source of biodiesel (Table A1). *Hermetia illucens* is usually mass reared to obtain products for animal feed, but it is also reared to obtain biodiesel, bioactive compounds, and for organic waste treatment (Table A1). *Lucilia sericata* (Meigen) (Calliphoridae) is used in maggot therapy, organic waste treatment, and forensic research (Table A1). The family Tephritidae includes various species that are mass reared for SIT applications (e.g., *Ceratitis capitata* (Wiedemann), *Anastrepha ludens* (Loew), *A. obliqua* (Macquart), *A. suspensa* (Loew), *Bactrocera cucurbitae* (Coquillett), *B. tryoni* (Froggatt)) [25,56,57,58] but also include species that are reared as experimental subjects or for animal feed and human food (Table A1).

Although we do not include a specific section dedicated to the mass rearing of flies for SIT applications, we note that the SIT is one of the most environmentally friendly and effective techniques to control Diptera of economic importance worldwide [25,57,59]. The SIT contributes to reducing the use of insecticides that are harmful to the environment and human health [60]. Of the more than 350 arthropod species of economic importance that have been targeted for radiation biology studies, about 30% are Diptera [57]. Among these, the most numerous are Tephritidae (41 species) [57].

The larval diets reported in Table A1 range from plain natural diets, such as carrion used for necrophagous species (e.g., *Chrysomya chloropyga* (Wiedemann) (Calliphoridae)) or living insect host in the case of tachinid parasitoids, to complex chemically defined (i.e., holidic) artificial diet formulations such as those used for rearing *D. melanogaster*, one of the most widely used model organisms in many medical and biological fields (Table A1). Rearing on an artificial diet is especially important for calliphorid species used in maggot therapy (Table A1) to ensure that the larvae to be applied to the wounds of patients come from a highly controlled and sterile rearing environment [61,62]. The families Calliphoridae, Muscidae, Sarcophagidae, and Stratiomyidae include species whose larvae are used for waste bioconversion and treatment of organic wastes such as poultry waste, fish waste, carrion, meat, animal manure, kitchen waste, and rotting plant material, among others [22,63] (Table A1). The larvae of the following species stand out for the bioconversion of waste materials: *H. illucens* for manure and plant waste; *M. domestica* for manure; *C. chloropyga, L. sericata* and *Sarcophaga dux* Thomson (Sarcophagidae) for carrion; *C. megacephala and Chrysomya putoria* (Wied.) for both carrion and manure; and *Musca autumnalis* De Geer for cattle manure; among other substrates [22].

Perhaps one of the most outstanding cases of successful rearing flies is that of sarcophagid parasitoids of lepidopteran pests such as *Sarcophaga aldrichi* Parker (Table A1), which can be continuously reared on artificial media lacking insect-derived material because females are ovoviviparous and lay larvae directly into the artificial medium [64]. Unlike the rearing of sarcophagid parasitoids, rearing tachinid parasitoids on artificial media involves collecting eggs or larvae from previously parasitized hosts and transferring them into the rearing medium [64].

We found that many studies lack specific details on adult diet and that relatively few studies have focused on adult diets in comparison to the body of research on larval diets (but see [31,48,61,65,66,67]). The adult diet may be irrelevant for species that use the storage of nutrients acquired during the larval stage for egg development such as *H. illucens* [68]. Although adult *H. illucens* do not require food for mating and oviposition [63], providing flies with water and sugar increases their longevity [69]. In contrast with *H. illucens*, the rearing of the calliphorids *L. sericata* and *Phormia regina* Meigen (Calliphoridae), and the muscids *Haematobia irritans* (L.) and *S. calcitrans*, involves adult diets containing blood alone or combined with other ingredients such as sugar and an alkalinizing agent to promote oviposition (Table A1).

In the following sections, we will address in more detail the topics of rearing flies for animal and human food, pollination services and pest control, medicinal wound therapy treatment, and forensic research and will highlight the relevance of laboratory-reared *D. melanogaster* as a vehicle of great scientific discoveries that have shaped the human understanding of many biological systems.

## 4. Rearing Flies for Animal Feed and to Feed the Growing Human Population

It is estimated that the human population will grow by about 40% by the year 2100, reaching the staggering number of 11 billion humans on the planet [21]. This implies a greater demand for land, water, food, and an overall increased exploitation of resources [21]. Increasing food production to meet increasingly demanding consumers requires new technologies and best practices for the sustainable production of foods with high nutritional value [19]. Insects are rich in high-quality nutrients [70,71,72], and their large-scale production has less environmental impact than traditional livestock production as a source of protein, requiring relatively little space and water consumption [73,74]. Insects have short life cycles, and some species can recycle organic waste, transforming them into high nutritional value biomass sources [21]. These characteristics have made *H. illucens* and *M. domestica* the subjects of intensive rearing in several places worldwide as two of the most promising insect species for industrial feed production [21,73,75,76], and, in the case of *H. illucens,* it is also a candidate for food applications [75].

The larvae and prepupae of *H. illucens* and *M. domestica* are rich in high-quality protein and lipids [77]. The meal from *H. illucens* and *M. domestica* can replace soymeal and fishmeal in the diets of poultry, pigs, fish species such as the rainbow trout and salmon, and ruminants [21,77]. *Hermetia illucens* larval meal provide high-quality, easily digestible protein in dry dog food formulations [78,79]. The larvae and prepupae of *H. illucens* can be used alive or dehydrated to feed chicks, alligators, frogs, and lizards [69].

For fly feed products to enter the European and US markets, the following substrates for fly rearing must be avoided: ruminant proteins, kitchen waste, or old food containing meat and fish, manure, meat and bone meal, intestinal contents, and fecal sludge [80,81]. Rearing substrates having, for example, heavy metals or antinutritional factors must also be avoided as they could limit insect growth or be accumulated in the insect tissue and passed into the food chain [81]. Substrates for fly farming require strict control of contaminants to ensure safety standards throughout the value chain [81]. The following substrates can be used to produce insects for animal feed: commercial animal feed; by-products of crops; or the remains of fruits and vegetables discarded from the markets, food produced for human consumption but no longer destined for human consumption because it expired or due to manufacturing problems or packaging defects [80,81].

Kenis and colleagues [82] extensively reviewed small-scale production systems for *H. illucens* and *M. domestica* and separated the production systems into the categories of “Natural oviposition” and “Adult Rearing and Egg Production”. They compared several parameters that affect production systems and reported that colony maintenance of *M. domestica* is time consuming and demands expensive adult food (e.g., milk powder, sugar, egg) to achieve high oviposition rates. Whereas, in the case of *H. illucens*, adults do not feed; thus, no investment is needed in adult food [82]. More larval substrate types can be used in *H. illucens* than in *M. domestica,* but the meal from *M. domestica* contains higher levels of unsaturated fatty acids and crude protein than the meal from *H. illucens* [77,82]. Interestingly, the levels of essential biomolecules such as fatty acids and micronutrients, such as Ca in the meal from *H. illucens* and *M. domestica*, can be enhanced by the manipulation of the larval diet [77]. In *H. illucens*, including banana peels into the larval diet can increase the levels of P, K, Ca, Na, Mn, and Mg in larvae [83]. Additionally, Borel and colleagues [84] showed that *H. illucens* larvae could accumulate vitamin A precursors from their diet when it included foods rich in provitamin A carotenoids such as orange carrots, Cinderella pumpkins, sweet potatoes, and clementines. This is important to consider because the rearing of *H. illucens* could be a sustainable way of recycling nutrients and restoring them back into the food cycle [84].

The practice of rearing flies for human food is limited by cultural conventions and health concerns compared to rearing for animal feed. An exemplary case of fly rearing for human consumption is represented by the tephritid fruit fly *C. capitata*, which is mass reared on artificial diet for SIT applications in many countries, but an Israeli company uses the larvae to produce and commercialize high-quality protein powder, reduced-fat powder, dry larvae, and larvae oil as feed and food ingredients [85,86]. The larvae of *A. ludens* are rich in lipids and protein with a high content of essential amino acids and can increase the nutritional value of foods such as ice cream without affecting their organoleptic properties [87]. Another fly species with potential for human food is *H. illucens*, whose larval flour can be added to baked goods to improve their nutritional value [88]. Nutrients from *H. illucens* are a potential and valuable resource to cover the needs for protein and other essential nutrients in developing countries and for the growing human population [89]. Obstacles to including *H. illucens* in the human diet include the social stigma and legal prohibitions against the consumption of insects that develop on organic waste [89]. However, these obstacles could be overcome by using synthetic mass-rearing diets [90]. Despite the current limited use of flies as a food ingredient, the contribution of reared flies to food security is relevant because fly products can be used to feed various species of farm animals that are part of our daily diet.

The rearing of flies to produce nutritive products for animal feed or human food represents a paradigm in the circular economy model that proposes social and economic growth while contributing to protecting the environment [21,91]. Surprisingly, of the more than 30 fly species that are recorded as edible worldwide [92], rearing protocols are only known for a few of them. As it occurs with most edible insect species, edible flies are mostly harvested from the wild [75], an activity that threatens the subsistence of natural populations and is not sustainable as a significant source of food biomass.

### Environmental Concerns of Industrial Insect Rearing for Food and Feed

It is widely accepted that the large-scale production of insects as food and feed has major environmental advantages compared to conventional livestock production, including less use of land and water, lower greenhouse gas emissions, and the ability to transform large quantities of organic residues into high-quality products, among others [93]. However, since industrial insect rearing as food and feed is a relatively new concept, concerns have been raised regarding its environmental sustainability, and the number of studies quantifying the environmental impact of this activity has significantly grown during the last decade [94].

Evaluations of greenhouse gas and ammonia direct emissions during the rearing of *H. illucens* on an industrial-production substrate (a mixture of yeast concentrate from wheat and a starch-rich by-product from the wheat and potato industry, together with a binding agent) revealed that about 17 g CO_2_ equivalents were emitted to produce 1 kg of dry larval biomass [90]. A key point to consider is that most greenhouse gas emissions come from bacteria and fungi in the rearing substrate rather than from the insect itself in rearing saprophagous flies such as *H. illucens* and *M. domestica* [94,95]. Therefore, the reduction in greenhouse gas emissions in the rearing systems of flies used for bioconversion should consider modifications of the larval substrate composition without negative effects on other parameters such as insect yields and quality.

The introduction of non-native edible insect species in areas without regulatory entry and monitoring guidelines, and the establishment of facilities without adequate biosecurity measures, represents a great risk of biological invasion with potentially catastrophic consequences because of the lack of natural enemies that could regulate the populations of alien insect species [96]. In the case of *H. illucens* and *M. domestica*, although they are species distributed throughout the world, caution should be taken, as differences in biotypes of insects might pose a risk of inadvertent introduction of microbes such as *Wolbachia* and various other bacteria [25]. Caution cannot be waived on this issue in the case of new fly species emerging as candidates for factory farming in the future.

## 5. Rearing for Pollination Services and Pest Control

A search of the *Web of Science* under the keywords “Diptera”, “pollination”, and “rearing” revealed 28 papers published over the past 20 years. Most of these papers treated rearing as a secondary or incidental feature of the papers’ purpose, but several papers contained valuable rearing information. For example, Francuski et al. [97] provided a comprehensive rearing regime for the syrphid fly *Eristalis tenax* L. The authors provided valuable information about diet (Table A1); they included oviposition media details (soaked barley and other non-specified cereals), and they provided container conditions and environmental conditions (22–26 °C, 50–70% relative humidity and 12:12 L:D photoperiod). A very important value of this paper regarding rearing the specific target (*E. tenax*) was the information on genetic truncation that took place over the F4-F8 generations. In association with the papers on *E. tenax* rearing was the remarkable paper by Nicholas et al. [98], where details of the experiments were presented graphically, including a link to a video account of the rearing conditions. Both the written paper and the video contained information about rearing conditions that addressed all phases of the life history of *E. tenax*, and many of these techniques would apply to other insects, including many Diptera. For example, Nicholas et al. [98] presented techniques for quieting the flies and transferring them from cage to cage (the cages being standard sleeve containers known as “BugDorms”). It is also worth noting that Francuski et al. [97] used cereals in place of the rabbit dung used by Nicholas et al. [98] and was closer to the natural food of *E. tenax* larvae. It would be useful to see a comparison of the fitness of these larvae fed cereals vs. rabbit dung. All the studies mentioned thus far dealt with rearing syrphids for pollination and were based on species that were not predaceous as larvae.

Among the highly diverse ecological functions of Diptera, many species are important as pollinators, whereas others are excellent predators and parasitoids. Some species in the families Syrphidae and Asilidae can be both. For example, the robber flies (Asilidae) are fierce and voracious and may act as keystone predators in some natural and agricultural ecosystems [99,100,101]. Considering the literature on asilid flies, it becomes clear that the group is under-appreciated in their roles as shapers of natural and agricultural ecosystem stability. This is especially clear from the study by Wei et al. [99], who demonstrated the highly effective protection afforded by asilid larvae (*Promachus yesonicus* Bigot) against white grubs (Coleoptera: Scarabaeidae, including species of *Anomala* and *Holotrichia*) that damage the root systems of various crops. Wei et al. [99] provided details of the rearing of *P. yesonicus* with white grubs as larval prey. They reported the relative ease of rearing these asilids, with encouraging potential for mass rearing based on a system of mass reared white grubs. In the same paper discussed here, Wei et al. [99] demonstrated the efficacy of *P. yesonicus* as a predator of soil-inhabiting white grubs in a wheat cropping system. The paper by Wei et al. [99] is of exceptional value as it demonstrated a proof of principle that such fast-flying and voracious predators as robber flies can be managed in a rearing system, leading to confidence that other flies from Asilidae and from other families such as Tabanidae, midges (Cecidomyiidae), dance flies, and balloon flies (Empidoidea) can all be managed in mass rearing conditions, provided an adequate knowledge of the flies’ biology is understood. Unfortunately, very little is known of the biology of most of the predatory fly species. This makes it less feasible to rear these potentially excellent predators than was shown to be possible for *P. yesonicus*. Therefore, as most of these potentially beneficial flies are not adequately understood (despite their great potential in biological control programs), we confine our discussion here to two families that have received considerable attention in the insect rearing community: the syrphids and the tachinids. The syrphids are of special value, considering the dual nature of their life history: predators as larvae and nectar/pollen feeders as adults [102].

### 5.1. Diets for Predaceous Syrphids

Iwai et al. [103] provided a basic study of feeding syrphid larvae on artificial diet. The larval diets consisted of drone powder (from honeybees, *Apis mellifera* L.) (Table A1). The drone powder diets were made from powdered drones mixed with brewer’s yeast autolyzate and sucrose. The diets were presented in each of two forms: as agar suspensions or as “powdery” slurries adhering to a sponge. The authors specified that the agar in the suspensions was about 0.5%. They showed that diets with higher concentrations of agar (therefore, having greater gel strength) were less acceptable than diets with lower agar concentrations. Iwai et al. [103] performed experiments with the proportions of drone powder (presumably from *A. mellifera*) to determine near-optimal concentrations of this important ingredient, using 10 to 60% drone powder at 10% increments. They showed that the more drone powder used, up to 50%, the better the survival rate, though none of the diet formulations resulted in survival rates as high as those of fresh aphid controls.

### 5.2. Tachinid Rearing

A search of *Web of Science* under the keywords “tachinid” and “rearing” yielded 122 references, indicating that this is an active topic of research. The considerable attention to tachinids is a result of the excellent reputation that the Tachinidae hold in the biological control community. The tachinids are generally somewhere between fastidiously host-specific hymenopteran parasitoids and generalist predators in their host range. The degree of specialization in this family makes tachinids desirable candidates for augmentation biological control programs. While most papers in this search reported rearing of tachinids on natural hosts and factitious hosts, several papers dealt with artificial diets as a central focus (Table A1). We will cover a few outstanding reports on artificial diets, but factitious hosts are most promising and practical for any kind of applied outcome towards true mass rearing. As most of the papers that reported rearing tachinids on natural diets gave little detail on rearing procedures, we neglected to provide detailed reviews of papers reporting such outcomes. For factitious hosts (used in the following sense: “A host other than the target host for parasitoids, one that biocontrol practitioners may more readily rear than the target host in a laboratory, thus implying that the biocontrol agent is not absolutely monophagous”, [104]), one of the most common ones used throughout the rearing community was *Galleria mellonella* (L.) (Lepidoptera: Pyralidae). Gross et al. [105] and Gross [106] provided useful details of the rearing system of the important parasitoid *Archytas marmoratus* (Townsend), which was mass reared and tested in the field. Importantly, Gross et al. [105] demonstrated that superior diets for the waxworm (*G. mellonella*) host translated to producing higher fitness in the parasitoids. The original techniques for using waxworms as factitious hosts for various species of tachinids were derived from Bratti and Costantini [107], Coulibaly and Fanti [108], and, earlier, by King et al. [109]. However, despite the relative ease in using factitious hosts such as *G. mellonella* to rear tachinids, the ultimate potential in mass rearing parasitoids for biological control remained as a rearing system based on artificial diets. Directly rearing predators and parasitoids on an artificial diet eliminates the “middleman-” concept: rearing an insect to rear another insect.

Unfortunately, the goal of developing a practical artificial diet for parasitoids has been an elusive goal, especially for parasitoid rearing [104]. Several papers provide very detailed, informative, and useful information about using artificial diets and the technology that is necessary for artificial diet-based rearing (including special containers, diet presentation systems, and oviposition setups). The paper by Nettles et al. [110] provided a culmination of decades of research on the development of artificial diets for parasitoids, especially for tachinid flies. Nettles et al. [110] provided a table of the diet components, which included 72 ingredients, comprising a nearly defined (holidic) diet. The authors credited prior works by Simon Grenier, whose work spanned at least three decades of inquiry (from the 1970s to the 2010s) into the nuances of parasitoid feeding biology. The diet(s) presented in Nettles et al. [110] included a few complete proteins and/or peptides such as bovine serum albumin, lactalbumin hydrolyzate, Yeastolate, and Bactopeptone. However, most of the nitrogenous nutrients were free amino acids. The diet contained the vitamins known to be required by insects, all the bases for nucleic acids, all the micronutrients such as minerals, and trace lipids. The composition of the diet was arrived at from hundreds of trials of various components in numerous combinations. In addition to the nutritional complexity of the diet, the Nettles group also considered the issues of the liquid vs. gelled diet as methods of helping the larvae receive better access to air for gas exchange (for example, by using different concentrations of agar), and the group also paid careful attention to the osmotic pressure of the diet, realizing that too high or too low an osmolarity would cause the larvae either to lose water to the media or gain an excess of water by osmosis. Behind the research to develop this group of diets, the Nettles group used various methods to sterilize the diet such as filtration and heat, as well as using prophylactic antimicrobial substances that they (with influence from the Grenier group) tested against *Eucelatoria* sp. as their target. Of the various diet formulations that they reported in the Nettles et al. [110] paper, the highest rate of pupation that they achieved was 15%. This was clearly too low for practical mass rearing, but it represented incredibly thoughtful and thought-provoking inquiry, and it should set the stage (along with the works from the Grenier laboratory) for the development of a practical diet, especially one that is less defined and more tied to whole (undefined) components such as bird eggs and meat products from vertebrates (e.g., internal organs and muscle meats).

## 6. Rearing of Medicinal Maggots

### 6.1. Medicinal Maggots for Wound Therapy

The extent of human suffering that derives from chronic wounds is vast. Various conditions such as diabetes, burns, certain cancers, lacerations, and many other conditions often lead to wounds that do not heal over protracted periods [111,112]. Observations discussed here of wounds whose healing was expedited through maggot therapy (MT) or maggot debridement therapy (MDT) point to the potential benefit of a much deeper understanding of the medical potential that MT/MDT could have for reducing suffering. The rationale for using maggot therapy (efficacy, safety, and simplicity) offers compelling reasons for pursuing the production of flies that can be used for therapeutic purposes.

### 6.2. Rearing Practices for Fly Colony Maintenance in Maggot Therapy

The phrase “maggot therapy” yielded 559 publications between 1934 and July of 2022 in the *Web of Science*. Although most of these publications offer testimony affirming the remarkable value of maggot therapy, a few recent articles pointed out some caveats and limitations of this practice. In all cases, proper caution in applying this unique therapy could alleviate concerns about infection or other secondary problems. Masiero et al. [65] described the general adult food for calliphorids such as water, sugar, and ground beef, which promoted oviposition, and, after disinfection with 1% sodium hypochlorite (NaClO), the eggs were placed on larval diets.

The simplest and most complete source of rearing instructions were found in Sherman and Wyle [113], where the authors suggested rearing various species of flies, including *Phaenicia sericata* (now called *L. sericata*), *P. regina*, and *L. illustris* on a diet of chicken eggs.

Stadler [62] pointed out that there was a dearth of publications fully dedicated to the rearing of maggots for medicinal purposes. Stadler cited works of Baer [114] and McKeever [115] as the principal bodies of information on colony maintenance for maggot therapy. One of the important areas of research for production of MT flies is the development of artificial diets (e.g., [61,116]; Table A1). Formulated appropriately, artificial diets could offer MT rearing programs with a source of food that could be all at once nutritious, inexpensive, and capable of being sterilized to avoid or reduce microbial contamination, which could be very consequential in maggot therapy.

### 6.3. Benefits and Risks of Maggot Therapy

However, though many of the hundreds of papers on MT/MDT present enthusiastic and encouraging advocation of this practice, some recent papers such as Connelly et al. [117] and Bueide et al. [118] reported some of the possible risks of maggot therapy, including infections caused or encouraged by the fly larvae. Naik and Harding [119] provided an excellent review of the overall concept of maggot debridement therapy (MDT). One interesting facet of the Naik and Harding paper was the history of MDT in the context of human history. They explained that the maggots had played roles in wound healing for thousands of years, and the published recognition of the potential benefits of maggots was mentioned in several references by army surgeons, dating back to the 1500s and being chronicled by surgeons in several wars. Interestingly, despite observations of benefits to chronic wounds, some involving gangrenous or otherwise necrotic tissues, the surgeons did not make efforts to use maggots proactively by inoculating patients with maggots. In one case, Naik and Harding mentioned that a surgeon (Dominique Larrey (1766–1842)) could not convince the wounded soldiers (from Napoleon’s army) to allow him to apply maggots to their wounds, but rather risked the effects of necrosis. The Naik and Harding [119] review of the literature is valuable also because it included papers that reviewed the state of the art and science of MDT, citing papers that used meta-analyses of this most promising practice.

### 6.4. Mechanisms of Action by Maggots

Chambers et al. [120] and Masiero et al. [121] summarized three main mechanisms involved in wound healing: (1) debridement per se; (2) disinfection/antibacterial activity; and (3) stimulation of healing. These topics and sub-topics were also explored by Bexfield et al. [122,123], where the authors documented antibiotic activity. Masiero et al. [121] demonstrated very active bactericidal activity from *Cochliomyia macellaria* (Fabricius) (Calliphoridae) “exo-secretions” against two prominent infection-causing bacteria, *Pseudomonas aeruginosa* and *Staphylococcus aureus*.

Sherman et al. [124] wrote a comprehensive review of maggot therapy. They summarized the practice as follows: “Certain fly larvae can infest corpses or the wounds of live hosts. Those which are least invasive on live hosts have been used therapeutically, to remove dead tissue from wounds, and promote healing. This medicinal use of maggots is increasing around the world, due to its efficacy, safety and simplicity. Given our low cultural esteem for maggots, the increasing use and popularity of maggot therapy is evidence of its utility. Maggot therapy has successfully treated many types of chronic wounds, but much clinical and basic research is needed still”. In the review by Sherman et al. [124], the biology of myiasis and the history of maggot therapy were presented; the current status of our understanding and clinical use of medicinal maggots was discussed, and opportunities for future research and applications were proposed.

Recently, Stadler [62] reviewed the literature on maggot therapy; this included an update of the review by Sherman et al. [124,125]. Stadler’s paper dealt with issues of transport and handling of the maggots at the sites where they were to be applied. This was an important aspect of MDT as it dictated the quality of the maggots and their fitness to perform the important medical task expected of them. As with other situations where reared insects were used for human purposes, simple problems in rearing or post-rearing/transportation issues could render the insects useless. Stadler [62] pointed out that research is needed on the transport and distribution of medicinal maggots for more effective maggot therapy (MT), including the organization of supply chain protocols. He also suggested that more information was needed about patients’ attitudes towards being subjected to MT. Finally, Stadler pointed out that the post-treatment handling of maggots must be established, including the humane disposal of the living insects.

### 6.5. Commercial Suppliers and Medical Procedures

A crucial part of using medicinal maggots includes handling and application techniques. Even such a simple concept as disposal requires special precautions and procedures because of the nature of the living organisms and the reality of the biohazard potential associated with wounds and human/animal tissues. Due to spatial limitations, this review will not probe these issues deeply, but a search of the literature reveals abundant resources that come from medical practitioners such as three of the major suppliers of medicinal maggots:International Biotherapy Society (http://biotherapysociety.org/ accessed on 11 June 2023);BioMonde (https://biomonde.com accessed on 11 June 2023) in Germany (but they also operate in Great Britain);Monarch Labs (https://www.monarchlabs.com/ accessed on 11 June 2023).

Papers such as Wilson et al. [126] provided detailed instructions on handling (applying, treatment time, and disposal) of maggot delivery devices such as the BioBag from BioMonde (see above). Wilson et al. [126] provided 42 references that detailed and documented the various factors in wound treatment, including temperature, rates of debridement vs. numbers of larvae and their pre-treatment (such as state of hunger), relationships of protease inhibitors and various kinds of bacteria, and other relevant topics.

## 7. Rearing of Forensic Flies

Forensic entomology has a long history, dating back to the 13th century in China, where the culprit of a stabbing was discovered thanks to blowflies that were attracted to traces of blood imperceptible to the human eye on the murder weapon [127]. Forensic entomology is based on a large body of research on decomposition ecology and involves the study of insects recovered from human corpses to generate information useful in criminal investigations [128,129]. In a *Web of Science* search using the terms “forensic”, “insect”, and “rearing”, we found 105 results as of August 2022. Among the Diptera of forensic importance, Calliphoridae is a family notable for their applications to determine the postmortem interval of corpses [130,131,132,133] (Table A1). Larval diets for rearing calliphorid species can be as simple as an animal protein source such as chicken or beef liver, to more complex semi-synthetic formulations, including agar, sardine, preservatives, and various other ingredients (Table A1). Adults can be fed with water, sugar, and blood or liver to promote oviposition, as in *L. sericata* and *P. regina* (Table A1). The Muscidae also include necrophagous species of forensic importance such as *Hydrotaea aenescens* (Wiedemann) and *H. spinigera* Stein (Table A1).

A paper that is a useful model of rearing insects to be used for forensic purposes is by Gruszka and Matuszewski [134]. Though this paper was not about Diptera, it is included here because of its value as a template for presentation of rearing protocols for insects of forensic importance. Though Sherman and Tran [135] expressed the purpose of their paper to include myiasis (including maggot therapy, covered elsewhere in the present paper), their methods are useful as models of rearing Diptera of forensic importance. The precautions and meticulousness of the rearing conditions are of utmost importance to the use of dipterans and other forensically important insects for proper interpretation of forensic analysis in projecting time of death, for example. Weidner et al. [136] provided another exemplary paper that should be used as a template for forensic rearing techniques. Their statement, “A lack of knowledge in collection techniques and limited access to an appropriate food source are the main reasons for absence in adequate collection and rearing protocols”, makes the point about the need for meticulous adherence to all rearing control procedures, including time, temperature, population density, and other related factors.

The subjects of the Weidner et al. [136] study were field fresh (recently collected) calliphorids *P. regina* and *C. macellaria*: two commonly found fly species of great forensic value. Chaudhury and Skoda [137] provided information on rearing *C. macellaria*. These procedures are also potentially useful for several other flesh-eating flies. The diets they tested consisted of fresh or dried bovine blood and liver diet (Table A1). They showed good efficacy of the blood-based diets, and their rearing conditions reflected the meticulous adherence to standard techniques, which served as templates for forensic models. An interesting problem with forensic insects was pointed out by Zuha et al. [138], demonstrating that certain phorid flies (*Megaselia scalaris* (Loew)) could enter the colony and contaminate it. Estrada et al. [139] used artificial diets to show that certain toxins and drugs that may occur in the corpses of crime victims could inhibit the rate of development of *Chrysomya albiceps* (Calliphoridae), rendering the calculations of time and manner of death questionable. In *L. sericata*, heavy metals such as cadmium, zinc, and copper can affect immature development and obscure the estimation of postmortem intervals [140]. Several other papers in the current search revealed similar findings about various substances, including ethanol, which have inhibitory effects on the development of *P. regina* [141].

A critical factor that affects the development of forensically important flies and that could affect estimations of the postmortem interval is temperature [142,143,144]. In this regard, Grassberger and Reiter [142] proposed the use of developmental models such as isomegalen and isomorphen diagrams to facilitate rapid and more precise estimates of the postmortem interval.

## 8. Laboratory-Reared *Drosophila melanogaster* as a Model for Great Scientific Discoveries

The vinegar fly *D. melanogaster* is, arguably, the most widely reared fly species that has been the vehicle of paramount scientific discoveries, starting with the experimental work of Thomas Hunt Morgan, who established the chromosomal theory of heredity winning the Nobel prize in Physiology or Medicine in 1933 [145,146,147]. Other brilliant scientists, including Hermann Muller; George Beadle and Edward Tatum; Max Delbrück, Alfred Hershey and Salvador Luria; Edward B. Lewis, Christiane Nüsslein-Volhard and Eric F. Wieschaus; Richard Axel and Linda Buck; Bruce A. Beutler, Jules A. Hoffmann and Ralph M. Steinman; and Jeffrey C. Hall, Michael Rosbash and Michael W. Young also won the Nobel prize for their discoveries in the fields of X-ray mutations, gene activity, replication and structure of viruses, development, olfaction, immunity and circadian rhythms, respectively, conducting research entirely or in part with laboratory-reared *D. melanogaster* [147]. The contribution of laboratory-reared *D. melanogaster* to the development of biology has been so important that it has been said that this fly species has a symbiotic relationship with humans, in which “food and habitat is traded for biological insights of remarkable breadth” [148]. To put it simply, much of the current knowledge in many biological and medical fields and the way we understand many biological systems are based on studies performed with laboratory-reared *D. melanogaster* as a model organism [145,146,149,150].

Many genes that have been found in *D. melanogaster* and are defined as being relevant for fly development and survival are also important for the development of other organisms, including humans [146,150]. Several biological mechanisms and pathways are conserved across evolution between *D. melanogaster* and humans [146], including most of the human disease genes [147,151]. Notably, laboratory-reared *D. melanogaster* has been used as an animal model to better understand the cellular steps and mechanisms leading to human cancer [152,153]. The genetic and environmental factors of human-related neurodegenerative diseases such as Alzheimer’s [154,155] and Parkinson’s [156] have also been modeled using experimental *D. melanogaster* individuals. Likewise, our understanding of the mechanisms underlying chronic diseases such as obesity and diabetes [157,158,159], and aging in relation to diet [147,160], have been much improved using laboratory-reared *D. melanogaster.* The hope that in the future there will be a cure for such terrible diseases that cause so much pain and suffering to millions of people worldwide will be supported by using *D. melanogaster* as a model organism.

## 9. Key Topics of Research in Fly Rearing

Studies on the basic aspects of life history, feeding behavior, nutritional physiology, and ecology are required to develop and optimize the rearing practices of many fly species [64,75,102]. In the case of dipteran parasitoids, the feeding mechanisms and nutritional needs of adults are poorly understood, and this needs to be further explored to develop large-scale rearing protocols for biocontrol programs [64]. Current research needs in Diptera-rearing include the development of cost-effective artificial diets, which is especially important for flies used in maggot therapy [61,62], in biological control programs against insect pests [102], and for SIT applications [56,161,162,163]. The development of cryopreservation techniques for larvae reared for maggot therapy is relevant because these larvae are very perishable, and commercial producers must ensure that maggots are delivered in perfect condition and in a short time [62].

A topic of vast importance in the specific context of Diptera rearing and in the general context of Diptera evolution is the intricate and intimate associations of flies and various microbes [24,25,164,165]. From the first rearing efforts with dipteran subjects, there was a recognition that flies and microbes were closely associated, and there was extensive dependence on microbes by various fly species [26,27,28,35,38,43]. One of the most insightful statements about this association was made by Bridges and Darby [38], where the authors considered the various names of *D. melanogaster* (vinegar fly, pomace fly, fruit fly, and several others), and these authors concluded that the epithet “yeast fly” would be the most fitting. In keeping with these early observations, dozens of papers were written about various associations of flies and microbes, especially interactions and interdependencies by drosophilids with tephritids and various yeasts (e.g., [166,167,168,169,170,171,172]). Investigations of the various kinds of associations have been spotty, with more than 200 references (*Web of Science* search on 19 December 2022) for “endosymbionts and Diptera” vs. only a few publications on “Diptera and ectosymbionts”, “Diptera and mycytomes/myceotcytes”, or “Diptera and mycangia”. A few research groups such as those from several laboratories at the University of Arizona (e.g., [166,167]) have delved deeply into the reciprocal community shaping effects of the *Drosophila* species and various yeast species. These studies are elegant and most informative about the interdependence of certain microbes (especially yeasts) and their dipteran hosts/associates [170], but they seem to have only scratched the surface of the relationships between Diptera and microbes. This is especially true in the applications of basic symbiotic relationships and their implications for mass rearing members of Diptera. However, we recognize significant research efforts to explore the relationships between gut microbes and flies to develop better artificial larval diets based on bacterial strains for mass rearing drosophilids and tephritids [56,164,173].

In addition to microbes, other factors in the larval diet, such as the nutrient composition of food and the number of individuals feeding on the same food resource, are known to influence life history traits that are of utmost importance for rearing [174,175,176]. A fascinating topic of research that could help establish and improve the rearing practices of fly colonies is that of thermotolerance induced by the larval density in the diet [177]. Regardless of the purpose of rearing, understanding diet-induced phenotypic variation is critical for successful rearing programs.

Another topic of research with practical applications in mass rearing is that of the loss of genetic diversity of reared flies due to genetic drift, inbreeding, and founder effects [97,178]. As such, researchers and managers of rearing facilities should be aware that captive populations of flies should be refreshed with wild material to ensure high levels of genetic and phenotypic diversity [97] and to prevent inbreeding [178]. Of course, any efforts to refresh or augment genetic diversity must be tempered with careful consideration of microbial disruption that often accompanies the addition of field-derived individuals [25]. Certainly, the management of colonies in rearing systems involves consideration of all the biotic and abiotic factors that could affect the outcome of the rearing and the quality of the reared insects [25,56].

## 10. Conclusions

The long and illustrious history of the science and technology of fly rearing has contributed tremendously to our understanding of many fields of knowledge, including genetics, SIT, and quality control. The vast technological developments of fly rearing should stand as a reference point for many other insect rearing systems. As a human activity, fly rearing contributes to many aspects of our daily lives, for example, in the production of human food and feed products, in medicinal treatments of difficult-to-heal wounds, in transforming large quantities of organic wastes that we generate every day into useful products for the cosmetic or animal feed industries, in pollinating crops and protecting them from pests (thereby avoiding the use of pesticides), and even as evidence in criminal cases in the case of forensic insects. The progress of fly rearing will require the involvement of current and future generations of scientists and technologists passionate to find innovative methods of using flies to improve existing and emerging problems to humanity.

## Figures and Tables

**Figure 1 insects-14-00553-f001:**
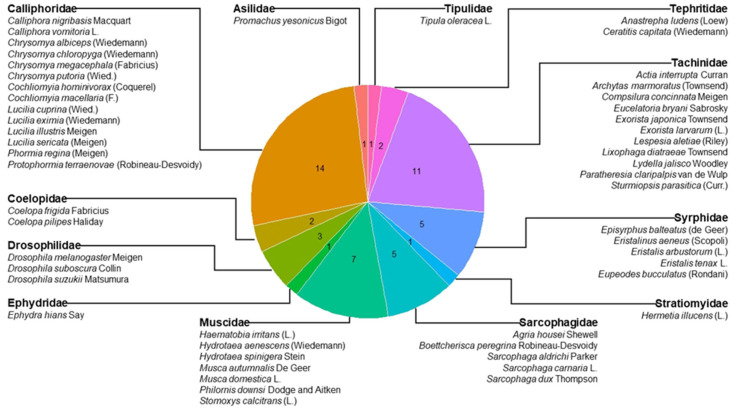
Pie chart of the fly species (N = 53) from 12 families whose uses and rearing diets are reported in Table A1. Numbers inside each slice indicate the number of species.

## Data Availability

Data are contained within the article.

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
