# Peer review of "How Rearing Systems for Various Species of Flies Benefit Humanity"

_insects, 2023, doi:10.3390/insects14060553_

Round 1

Reviewer 1 Report

This manuscript summarized and described well the information on the rearing of flies in the fields of the animal feed and human food industries, pollination services, pest control, medical wound therapy treatments, criminal investigations, and as model organisms for the development of several biological and medical fields, as well as the importance for the benefit of Humanity. I think this manuscript is acceptable in the current form. I have minor comments below.

1. The title may be changed to include the whole scope of the text, since this review is dealing with various topics for fly species in the benefit of Humanity.

Fly species for the benefit of Humanity: With special emphasis on the rearing systems

2. Table A1 is too long extending to 10 pages.

The column for Family can be removed by moving to just above the corresponding species. Also, the column for References can be simplified by referring number with the footnote below Table. Then, the column for Larval diets can be widened, which includes more information.

Reviewer 2 Report

(line)

1.

(9) → "generally...annoying little creatures, pests, diseases" → not in this journal; here, everybody has deep respect concerning insects; pls adapt the "tone". i understand your jocular remark but pls do not do it here.

(11) etc. → "Earth" may better be written small (earth), maybe ask for editorial decision

(46) → "pollinators": yes, indeed. pls insert a recent reference, this is currently a very important topic; you will find several references here → https://www.cell.com/current-biology/pdf/S0960-9822(22)00423-7.pdf

(108) → "set the stage": maybe a little too "chatty" style, pls tone down

(376) → maybe better italics for Web of Science 

(550, 551) → please delete or strongly shorten the "empty" sentence (sounds like a lecture not a scientific paper)

(559) → it is very unusual to "validate" a paper, so pls. remove "excellent"; it is clear that you like the paper because you quote it

(571) → same as (559); pls. remove "valuable"; this is not a report to an agency but a scientific paper 

(591) → biomonde is originally a german company; they also operate in other countries, especially england, but are genuinely german, the central offices are at BioMonde GmbH, Kiebitzhörn 33, Barsbüttel 22885 Deutschland = Germany 

2.

you found very many good references but oversaw some essential papers, please do quote them:

a.

https://pubmed.ncbi.nlm.nih.gov/11457606/

b.

MARCHENKO M.J. (2001): Medicolegal relevance of cadaver entomofauna for the determination of time since death. — Forensic Science International 120: 89-109.

c.

GRASSBERGER M. & C. FRANK (2003): Temperature-related development of the parasitoid wasp Nasonia vitripennis as forensic indicator. — Medical and Veterinary Entomology 17: 257- 262.

d.

GRASSBERGER M. & C. REITER (2002): Effect of temperature on development of Liopygia (= Sarcophaga) argyrostoma (Robineau-Desvoidy) (Diptera: Sarcophagidae) and its forensic implications. — Journal of Forensic Sciences 47: 1332-1336.

e.

(194)

conc. maggot therapy:

https://shop.thieme.de/Maggot-Therapy/9783132578173

it is relavant because it made it into a book publication in two languages

f.

(205) (599) conc. forensic applications → the following is the / a complete historical overview, pls. include it:

https://www.researchgate.net/publication/270758147_A_brief_survey_of_the_history_of_forensic_entomology_Ein_kurzer_Streifzug_durch_die_Geschichte_der_forensischen_Entomologie

g.

(475) conc. holidic medium for d. melanogaster:

https://www.nature.com/articles/nmeth.2731

language is fine

Author Response

Response to Reviewer 2 Comments

Point 1: (9) → "generally...annoying little creatures, pests, diseases" → not in this journal; here, everybody has deep respect concerning insects; pls adapt the "tone". i understand your jocular remark but pls do not do it here.

Response/action taken: We by no means were trying to be funny or intended to make readers laugh as Reviewer 2 implies (i.e., make a jocular comment).  Much less do we intend to disrespect readers as Reviewer 2 seems to have felt.  Not at all!

We feel that the tone of this comment of Reviewer 2 and the adjective “jocular” she/he uses is a bit condescending and does not contribute to the constructive criticism expected in the peer-review process of academic articles.  As indicated in the Microsoft Word template of Insects, the simple summary should describe the “work simply and concisely to the public” and “be written for a lay audience”.  And certainly, the general public/lay audience often has the view of flies as annoying little creatures that transmit diseases or are pests of crop plants.  However, considering that other readers may feel offended by the opening sentence of the simple summary as did Reviewer 2, we made the appropriate modifications to “adapt the tone” as requested (lines 10-12).

Point 2: (11) etc. → "Earth" may better be written small (earth), maybe ask for editorial decision

Response/action taken: “Earth” stands for the name of the planet and should be written with an initial capital letter.  No change was made.

Point 3: (46) → "pollinators": yes, indeed. pls insert a recent reference, this is currently a very important topic; you will find several references here → https://www.cell.com/current-biology/pdf/S0960-9822(22)00423-7.pdf

Response/action taken: The new reference was included as requested (line 48).

Point 4: (108) → "set the stage": maybe a little too "chatty" style, pls tone down

Response/action taken: We do not like Reviewer’s 2 use of “chatty,” but we changed the text to a more formal (and stuffy) expression and now it reads “Soon after Bogdanow established the use of artificial diet…” (lines 112-113).

Point 5: (376) → maybe better italics for Web of Science

Response/action taken: “Web of Science” is now written in italics as requested (lines 385, 449, 521, 616, 731).

Point 6: (550, 551) → please delete or strongly shorten the "empty" sentence (sounds like a lecture not a scientific paper)

Response/action taken: Change made as requested (lines 560-562).

Point 7: (559) → it is very unusual to "validate" a paper, so pls. remove "excellent"; it is clear that you like the paper because you quote it

Response/action taken: We disagree with Reviewer’s 2 statement that it is unusual for authors to “validate”/evaluate a paper that they are citing. We (as authors reviewing previous works) are knowledgeable about the background papers, and our comment that a paper is excellent does a service for the audience by letting them know that a paper is especially helpful, insightful, or otherwise worth reading.  But to avoid a sterile discussion with Reviewer 2 that could delay the publication of our work, we accept the comment and made the required change (line 569).

Point 8: (571) → same as (559); pls. remove "valuable"; this is not a report to an agency but a scientific paper

Response/action taken: Change made as requested (line 581).

Point 9: (591) → biomonde is originally a german company; they also operate in other countries, especially england, but are genuinely german, the central offices are at BioMonde GmbH, Kiebitzhörn 33, Barsbüttel 22885 Deutschland = Germany

Response/action taken: Correction was made (lines 601-602).

Point 10: you found very many good references but oversaw some essential papers, please do quote them:

a.

https://pubmed.ncbi.nlm.nih.gov/11457606/

Response/action taken: The paper by Grassberger & Reiter (2001) is now cited in lines 660-661.

Point 11: b.

MARCHENKO M.J. (2001): Medicolegal relevance of cadaver entomofauna for the determination of time since death. — Forensic Science International 120: 89-109.

Response/action taken: The paper was cited as requested (lines 660).

Point 12: c.

GRASSBERGER M. & C. FRANK (2003): Temperature-related development of the parasitoid wasp Nasonia vitripennis as forensic indicator. — Medical and Veterinary Entomology 17: 257- 262.

Response/action taken: This paper is outside the focus of our work as it is about a hymenopteran parasitoid.  No change was made.

Point 13: d.

GRASSBERGER M. & C. REITER (2002): Effect of temperature on development of Liopygia (= Sarcophaga) argyrostoma (Robineau-Desvoidy) (Diptera: Sarcophagidae) and its forensic implications. — Journal of Forensic Sciences 47: 1332-1336.

Response/action taken: The paper was cited as requested (lines 660).

Point 14: e.

(194)

conc. maggot therapy:

https://shop.thieme.de/Maggot-Therapy/9783132578173

it is relavant because it made it into a book publication in two languages

Response/action taken: We are unsure about what, exactly, Reviewer 2 means with “conc.” here and in the following comments.  We were unable to get this book.  No change was made.

Point 15: f.

(205) (599) conc. forensic applications → the following is the / a complete historical overview, pls. include it:

https://www.researchgate.net/publication/270758147_A_brief_survey_of_the_history_of_forensic_entomology_Ein_kurzer_Streifzug_durch_die_Geschichte_der_forensischen_Entomologie

Response/action taken: Reference was included as requested (lines 611-613).

Point 16: g.

(475) conc. holidic medium for d. melanogaster:

https://www.nature.com/articles/nmeth.2731

Response/action taken: The paper is already cited in our work in Table A1 (reference 186).

Reviewer 3 Report

I'm not a specialist in fly rearing, but for me the paper was fascinating to read and I learned a lot. 

Congratulations on this review.  In my opinion, the article can be published as it is. I just noticed a typing error on line 234: Thomson instead of Thopmson.

Author Response

Response to Reviewer 3 Comments

Point 1: I'm not a specialist in fly rearing, but for me the paper was fascinating to read and I learned a lot.

Response/action taken:  We thank Reviewer 3 for her/his positive feedback and feel encouraged to hear that she/he enjoyed reading our work!

Point 2: Congratulations on this review.  In my opinion, the article can be published as it is. I just noticed a typing error on line 234: Thomson instead of Thopmson.

Response/action taken: The typing error was corrected in line 241 and Table A1.